# Are Intravesical Aminoglycosides the New Gold Standard in the Management of Refractory Urinary Tract Infection: A Systematic Review of Literature

**DOI:** 10.3390/jcm11195703

**Published:** 2022-09-27

**Authors:** Andrea Ong, Amelia Pietropaolo, George Brown, Bhaskar K. Somani

**Affiliations:** University Hospital Southampton NHS Foundation Trust, Tremona Road Southampton, Hampshire SO16 6YD, UK

**Keywords:** urinary tract infections, multiresistant infections, novel antimicrobial treatments, UTIs

## Abstract

Background: Antibiotic resistance in urinary pathogens is increasingly common, leading to rising cases of complicated urinary tract infections. Conventional antimicrobial treatment may be insufficient in these cases and broad-spectrum systemic antibiotics contribute to the problem. Intravesical aminoglycoside instillation is an alternative treatment option that delivers localized and high-dose treatment to the source of infection. This study summarizes the existing evidence for the efficacy and safety of this treatment. Methods: A systematic search was conducted of worldwide literature according to PRISMA methodology and Cochrane standards for systematic review. Studies were included if they reported outcome data for the prevention and reduction in urinary tract infections, eradication of antimicrobial-resistant organisms, or change in sensitivities allowing conventional oral antimicrobial treatment after the administration of intravesical aminoglycoside with or without polymyxin therapy. Results: The search identified 826 articles, of which, 19 were included in the final data analysis and narrative synthesis. A successful outcome was identified in 80.7% (*n* = 289) of patients treated with aminoglycoside alone and 79.5% (*n* = 163) treated with an aminoglycoside in combination with polymyxin. Discontinuation was noted in 6.2% of patients. An increase in antimicrobial sensitivity was seen in 15.3% (*n* = 55) and 16.3% (*n* = 36) in the aminoglycoside and aminoglycoside/polymyxin groups, respectively. Conclusions: Current evidence supports the use of intravesical aminoglycoside instillation as an efficacious and safe treatment for refractory UTIs. Nevertheless, data is limited, and larger volume studies with longer follow-up periods are required.

## 1. Introduction

The increasing incidence of multi-drug resistant organisms is one of the greatest challenges facing modern medicine. Urinary tract infections are one of the most common reasons for conventional antibiotic prescribing and resistance amongst urinary bacteria is an increasingly significant issue. The availability of effective alternative treatment strategies is therefore imperative, particularly in cases of complicated, refractory, and recurrent infection. One option is intravesical antimicrobial instillations. Existing literature indicates that this could be useful in the treatment and prophylaxis of UTIs, but its true role has yet to be fully established [1,2]. This review will focus on the efficacy of intravesical aminoglycoside instillations (IVA). Research to identify efficacious treatment options is vital in the effort to overcome the significant burden of UTIs on primary and secondary healthcare. Improved management of UTIs will improve the quality of life on an individual level and positively impact the global struggle against growing antimicrobial resistance [3].

## 2. Methodology

Studies for inclusion in this review were selected according to the following criteria.

Inclusion criteria:Studies reporting on the administration of intravesical aminoglycosides for the treatment and prevention of refractory UTIs.Articles written in the English language.All age groups, including pediatric studies.Studies with a minimal sample size of three patients.Exclusion criteria:Non-human studies, review articles, editorials, guidelines, and case reports.Studies with a non-UTI treatment indication.Studies reporting on non-aminoglycoside intravesical instillations.

### 2.1. Search Strategy and Study Selection 

A systematic review was conducted according to the Cochrane and preferred reporting items for systematic reviews and meta-analyses (PRISMA) standards [4]. An electronic search strategy was performed to find all relevant publications pertaining to intravesical aminoglycoside instillation for the treatment or prophylaxis of UTIs. Databases included AMED, CINAHL, British Nursing Index, Cochrane library, EMBASE, OvidEmcare, HMIC, Medline, PsycINFO, social practice & policy, and science direct/Scopus. References were cross-checked and individual urology journals were hand-searched.

Boolean operators (AND, OR) were used alongside keywords including ‘aminoglycosides’, ‘gentamicin’, ‘neomycin’, ‘nebramycin’, ‘intravesical’, ‘instillation’, ‘irrigation’, ‘recurrent’, ‘UTI’, and ‘urinary tract infection’. The literature search included English articles from inception to May 2022. Eligible studies identified by the search were screened by title and abstract and then by the full text to identify eligible studies.

### 2.2. Data Extraction and Outcomes of Interest

Data extraction was conducted independently by two authors (A.O. and A.P.) and discrepancies were resolved by mutual consensus. A decision was made to include articles investigating aminoglycosides in conjunction with polymyxin to allow for comparison against the efficacy of aminoglycosides alone. The measure of a successful outcome differed between the studies. Therefore, to aid comparison, this review defined a successful outcome as the prevention of UTI, a ≥50% reduction in the rate of UTI, eradication of antimicrobial-resistant organisms, or the change in sensitivities allowing for the oral antimicrobial treatment of UTI. Articles were reviewed for data on method of delivery, follow-up, change in antimicrobial sensitivity, discontinuation of treatment, and antibiotic dose and regime.

## 3. Results

An initial search identified 826 articles; 32 were selected from an analysis of the title and abstract for full-text review. In total, 3 were excluded as they were case reports with fewer than 3 participants, 2 were systematic reviews, 7 were irrelevant, and 1 was an animal study. After an analysis of the full text, 19 were included in our final review (Figure 1); 12 aminoglycoside and 7 aminoglycoside-polymyxin articles (Table 1 and Table 2).

A total of 584 patients were included in the systematic review. The mean age was 35 years. The gender demographics of participants were not included in two studies, and of those listed, 222 were male and 162 were female. In summary, 10 studies assessed the efficacy of gentamicin, 3 studies treated gentamicin-resistant patients with amikacin, tobramycin, or garamycin, and 2 studies assessed netilmicin and neomycin. Seven studies assessing the use of an intravesical aminoglycoside in combination with polymyxin were included for comparison. All the studies were observational, apart from two randomized control trials comprising 52 participants [17,20].

The indication for the use of IVA was a mixture of treatment and prophylaxis of refractory UTIs. Underlying pathology included intermittent self-catheterization/long-term catheterization, recurrent UTIs, prostate carcinoma, benign prostate hyperplasia, and urinary diversion. The most common underlying pathology identified was neurogenic bladder (listed in 74% of papers).

### 3.1. Clinical Efficacy of IVA

A successful outcome was identified in 80.7% (*n* = 289) of the patients in the papers assessing the use of aminoglycoside alone. A total of 79.5% (*n* = 163) of successful outcomes were measured in the papers evaluating treatment with an aminoglycoside in combination with polymyxin (Table 3 and Table 4).

### 3.2. Patient Compliance

The majority of the IVA administrations were undertaken by the patients themselves or their parents (63.2%, 12 out of 19 papers) [3,6,7,9,11,12,13,14,15,20,21,22]. The dosage of aminoglycoside (concentration of aminoglycoside, volume of aminoglycoside, and frequency of administration) varied greatly between studies. However, there was no significant association between dosage and discontinuation of treatment or development of new antimicrobial resistance. In total, 36 cases of discontinuation were noted (6.2%). Reasons for discontinuation included allergy, UTI, “clinical failure”, chronic prostatitis, “other health-related issues”, stones, pregnancy, chemotherapy, surgery (including cystectomy, nephrectomy, and bladder stone), and one case of newly developed gentamicin resistance.

### 3.3. Safety

An increase in antimicrobial sensitivity was observed in 15.3% (*n* = 55) and 16.3% (*n* = 36) in the aminoglycoside and aminoglycoside/polymyxin groups, respectively (Table 3 and Table 4).

Amongst these patients, oral antibiotics were used rather than intravenous antibiotics, which reduced UTI-related hospital admissions. New antimicrobial resistance was reported in 14.0% (*n* = 82) of patients; 10.4% (*n* = 61) in the aminoglycoside group and 3.6% (*n* = 21) in the aminoglycoside/polymyxin group.

One case of an elevated serum level of gentamicin was noted by Marei et al. [15]. No consequent side effects, including renal impairment or ototoxicity, were reported. Side effects were observed in 4.6% (*n* = 27) of patients. Renal impairment was reported in three patients. However, this was attributed to a known background of chronic renal insufficiency [9]. Stalenhoef et al. identified hearing loss in two patients [13]. Once again, this was attributed to pre-existing co-morbidities rather than aminoglycoside-induced ototoxicity. Other reported side effects included UTI, vaginal discomfort, yeast infection, diarrhea, allergic reaction, and autonomic dysreflexia.

## 4. Discussion

Localized, high-concentration IVA treatment aims to effectively eradicate multi-resistant organisms and limit growing antimicrobial resistance through the avoidance of broad-spectrum intravenous antibiotics [2]. This summary of the available literature supports existing evidence that the use of IVA is safe and successful in patients at risk of developing UTIs. Our definition of a successful outcome incorporated various parameters to allow for the comparison between different methodologies and measured outcomes. A successful outcome in the form of a reduction in UTI and/or reduced sensitivity allowing for oral antibiotic therapy was seen in both the aminoglycoside (82.1%) and aminoglycoside/polymyxin (79.5%) treatment groups.

Whilst the studies differ in duration of follow-up, findings consistently suggest that IVA has a role in the management of UTIs in the short term. Over a six-month period of review, Chernyak et al. found a statistically significant reduction in the incidence of UTIs from 2.5 to 1.5 per 6 months (*p* = 0.025) [14]. The study by Stalenhoef et al. explored a longer follow-up period of 42 weeks. While eradication of UTIs was observed in 41% of patients, 82% of the samples noted a 50% reduction in the rate of UTI, and only 8% of breakthrough UTIs required treatment with intravenous antibiotics [13]. Assessing a longer period of follow-up (an average of 26 months), Abrams et al. observed a reduction in UTIs in 22 out of 27 patients [3]. The reliability of these findings is limited by the relatively short periods of patient follow-up. Further research should aim to assess the long-term impact of IVA in terms of efficacy, safety, and patient satisfaction.

Complications such as ototoxicity and nephrotoxicity are well established with the use of high-dose aminoglycosides [23]. Intravesical administration aims to reduce drug toxicity by utilizing the tough impermeable nature of the urothelium to limit systemic absorption [24]. Despite the use of high concentrations of IVA, this review shows no evidence of systemic absorption of aminoglycoside. This suggests that a high-dose localized IVA can be safely used without systemic complications. However, these findings are limited as several of the studies did not include data on the serum level of aminoglycoside and associated side effects [5,6,11,14,16,17,18,19,20,21,22].

The low discontinuation rate of 6.2% and the high proportion of patients self-administering IVA suggests that patients were agreeable to this method of drug administration. Reducing the burden of staff required to administer the medication increases patient accessibility for IVA. Low complication rates (4.6%) and good patient compliance with the treatment suggest that IVA treatment is not only efficacious but a well-tolerated method of treatment for patients.

An improvement in antimicrobial sensitivity was observed from data collected from 10 papers. Although this was only a small percentage (15.3% (*n* = 55) and 16.3% (*n* = 36) in the aminoglycoside and aminoglycoside/polymyxin groups, respectively), this was a promising finding in the endeavor to reduce antimicrobial resistance. Increased antimicrobial susceptibility broadens the range of treatments available to these patients. This could avoid hospital admission, reduce the incidence of complications associated with hospital admission, and improve quality of life [3]. Chernyak et al. reported a statistically significant reduction in microorganism resistance from the median resistance of 8.5 antibiotics in a profile to 0 (*p* = 0.065) [14]. Overall, new antimicrobial resistance was reported in 14.0% (*n* = 82) of patients; 10.4% (*n* = 61) in the aminoglycoside group and 3.6% (*n* = 21) in the aminoglycoside/ polymyxin group. It was not possible to identify a trend between IVA and antimicrobial sensitivity based on our findings. Further studies should assess the long-term impact of IVA on antimicrobial resistance both in the treatment and the prophylaxis of UTIs.

### Limitations and Areas of Future Research

Most of the data included were retrospective and therefore vulnerable to bias and unable to be generalized across various healthcare settings. Most of the papers in the aminoglycoside group assessed the efficacy of gentamicin. However, the aminoglycoside used varied between studies and even within studies in some cases due to resistance. The reliability of the findings is therefore limited by inconsistent outcome measures between papers. For example, only 10 of the included papers included data on antimicrobial sensitivity, and only 8 papers reported on the serum level of aminoglycoside. Due to these data insufficiencies, definitive conclusions cannot be drawn. Additionally, although the majority of patients had a neurogenic bladder as the underlying pathology, there was great variation in the inclusion criterion between papers.

The review is limited by the heterogeneity of the studies and variation in sample sizes. Variability in study design was noted in the definition of recurrent/refractory UTI, the measure of successful outcomes, the categorization of groups into treatment versus prophylactic IVA, and the duration of follow-up. As a result of these key design differences, it was not possible to accurately evaluate the merits or failures of IVA treatment amongst these patients (Table 1 and Table 2).

The short follow-up in some studies prevents the true assessment of success in these patients. Five of thirteen papers are characterized by a short follow-up duration. Haldorson et al., Wan et al., Praba et al., and Cox and Waites et al. describe a follow-up that goes from a minimum of 4 days to a maximum of 8 weeks (6, 10, 12, 19, 24). Given that the definition of recurrent UTIs is ≥2 infections in six months or ≥3 infections in one year, UTI resolution should be defined with a longer follow-up after the treatment. However, in view of limited patient numbers and poor patient compliance to long-term follow-up, these papers were also included.

Future studies with a more robust methodology and longer follow-ups are required to draw meaningful conclusions. This would allow for a more accurate assessment of treatment efficacy with a view to aid clinical guidelines. It was noted the patient demographics were often not disclosed or sparsely documented in the included papers. This could be explored further to evaluate the optimum demographic characteristics of patients that would benefit from IVA and evaluate the impact of patient co-morbidity.

## 5. Conclusions

This study observed that IVA is an efficacious method of managing the treatment and prophylaxis of refractory UTIs in the short term. The intravesical method of administration allows for higher concentrations of aminoglycoside to be given in a localized approach with no systemic absorption of aminoglycoside or associated side effects. Patients were able to self-administer the treatment and the low discontinuation rates suggest that it is a well-tolerated treatment option. Further research with larger sample sizes, longer follow-up periods, and analysis of trends in antimicrobial susceptibility will better assess the efficacy of IVA for the treatment and prophylaxis of UTIs.

## Figures and Tables

**Figure 1 jcm-11-05703-f001:**
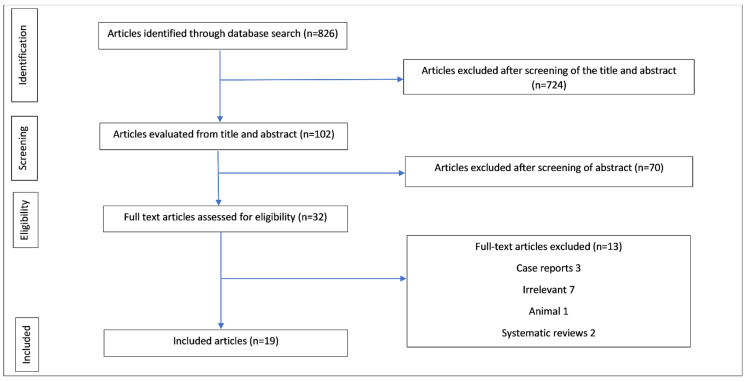
Diagram showing the PRISMA flowchart of studies searched, reviewed, and included in the systematic review.

**Table 1 jcm-11-05703-t001:** Characteristics of studies on the use of intravesical aminoglycoside for UTI.

Study	Study Design	No. of Patients	Mean Age	Male:Female	Additional Risk Factors/Comorbidities	Definition of Recurrent UTI in Included Studies
Haldorson et al., 1978 [5]	Prospective	53	ND	33:20	SCI *, multiple sclerosis, vascular disease, and cancer	Recurrent bacteriuria during intermittent catheterization
McGuire and Savastano 1987 [6]	Case series	4	68	0:4	Hemorrhagic cystitis, bladder dysfunction, and high residual volume	Failed response to oral antibiotic treatment
Wan et al., 1994 [7]	Prospective	10	1–18 yo	4:6	Myelomeningocele, vesicoureteric reflux, augmentation enterocystoplasty, Crohn’s disease, and renal transplant	Positive urine cultures
Arap et al., 2003 [8]	Retrospective	18	70	0:18	Recurrent UTI	Failed response to oral antibiotic treatment
Defoor et al., 2006 [9]	Retrospective	80	10	38:42	Neurogenic bladder, bladder exstrophy, cloacal anomalies, Hinman syndrome, vesicoureteral reflux, hypospadias, posterior urethral valves, bladder reconstruction, and renal transplantation	Failed response to oral antibiotic treatment
Praba and Utomo 2015 [10]	Prospective clinical trial	28	58.9	54:2	Benign prostatic hyperplasia and prostate carcinoma	Nosocomial catheter-associated UTIs
Abrams et al., 2017 [3]	Retrospective	27	55	7:20	Neobladders, neurogenic bladder, ileocystoplasty, and ISC **	Recurrent UTI, failed oral prophylaxis
Cox et al., 2017 [11]	Prospective	22	37.5	22:15	SCI, multiple sclerosis, and myelodysplasia transverse myelitis	Four UTIs in the preceding 6-month period
Dray VE et al., 2017 [12]	Retrospective	22	ND	ND	ISC	Three or greater urinary tract infections (UTIs) in one year, or two or more UTIs in six months
Stalenhoef et al., 2019 [13]	Retrospective	63	61	51:12	ISC, chronic bacterial prostatitis, vesicoureteral reflux, and neobladder	Recurrent UTIs despite failed oral prophylaxis
Chernyak and Salamon 2020 [14]	Retrospective case series	12	80.3	00:12	Neurogenic bladder and structurally abnormal urinary tract	Two UTIs in 6 months or three UTIs in a 1 year period
Marei et al., 2021 [15]	Retrospective	24	3.8	13:11	Neurogenic bladder, bladder exstrophy, cloacal anomalies, and posterior urethral valves	Recurrent UTIs despite failed oral prophylaxis

* Spinal cord injury (SCI). ** Intermittent self-catheterization (ISC). Not disclosed (ND).

**Table 2 jcm-11-05703-t002:** Characteristics of studies on the use of an intravesical aminoglycoside in combination with polymyxin for UTI.

Study	Study Design	No. of Patients	Mean Age	Male:Female	Additional Risk Factors/Comorbidities	Definition of Recurrent UTI
Pearman et al., 1988 [16]	Prospective randomized comparative	18	26	18:0	SCI	Consecutive positive urinary cultures
Pearman 1979 [17]	Randomized control trial	22	30	17:5	SCI	Consecutive positive urinary cultures
Linsenmeyer et al., 1998 [18]	Retrospective	12	ND	7:3	SCI	Culture-positive UTIs unable to be managed with oral antibiotics
Anderson 1980 [19]	Randomized prospective	17	ND	64:0	Acute neurogenic bladder	Significant bacteriuria
Waites et al., 2006 [20]	Randomized	30	ND	ND	SCI and neurogenic bladder	Recurrent microscopic bacteriuria and pyuria
Rhame and Perkash 1979 [21]	Retrospective	70	36.4	70:0	SCI	Positive urine culture
Huen et al., 2019 [22]	Retrospective	52	14.5	21:31	Spina bifida, cloacal exstrophy, posterior urethral valves, appendicovesicostomy, and enterocystoplasty	Frequent symptomatic UTIs despite oral antibiotic prophylaxis

**Table 3 jcm-11-05703-t003:** Administration, outcomes, and safety of use of intravesical aminoglycosides for UTI.

Study	Intervention	Follow-Up (Range)	Successful Outcome **	Discontinued Treatment	Side Effects	Serum Level	Change in Sensitivities	Developed New Resistance
Haldorson et al., 1978 [5]	0.1% neomycin solution	6 weeks	25/53	0	ND	ND	ND	37/53
McGuire and Savastano 1987 [6]	20 mL of 240 mg gentamicin in 1 L 0.9% NaCl *	20 months (3–36 months)	4/4	0	ND	ND	ND	ND
Wan et al., 1994 [7]	30–60 mL 480mg gentamicin in 1 L 0.9% NaCl	1 week	10/10	0	0	Yes, negligible	ND	ND
Arap et al., 2003 [8]	480 mg gentamicin in 1 L 0.9% NaCl + 100 mL sodium carbonate	65.1 months	12/18	3/18	3 UTI	Yes, negligible	0	3/18
Defoor et al., 2006 [9]	0.48 mg/mL gentamicin in 30 mL 0.9% NaCl	90 days	75/80	Variable not trackable	Minor rise in serum creatinine for three patients with chronic renal insufficiency	Yes, negligible	16/80	5/80
Praba and Utomo 2015 [10]	25 mg/1 mL netilmicin in 50 mL 0.9% NaCl	4 days	22/28	0	ND	ND	ND	ND
Abrams et al., 2017 [3]	80 mg gentamicin in 10 mL 0.9% NaCl	26 months (2–67)	22/27	6/27	0	Yes, negligible	18/27	1/27
Cox et al., 2017 [11]	14.4–28.8 mg gentamicin in 30–60 mL of 0.9% NaCl (according to bladder capacity)	6 weeks	22/22	0	1 yeast infection, 1 diarrhea	Not checked	9/22	8/22
Dray VE et al., 2017 [12]	14.4–28.8 mg gentamicin in 30–60 mL of 0.9% NaCl (according to bladder capacity)	ND	22/22	ND	0	Yes, negligible	ND	ND
Stalenhoef et al., 2019 [13]	80 mg gentamicin in 20 mL of 0.9% NaCl	42 weeks (6–148)	52/63	10	Hearing loss (*n* = 2), vaginal discomfort (*n* = 10), and abdominal discomfort (*n* = 3)	Yes, negligible	4/14	6/63
Chernyak and Salamon 2020 [14]	80 mg gentamicin in 60 mL 0.9% NaCl or 80 mg tobramycin in 100 mL 0.9% NaCl	6 months	12/12	0	0	Not checked	8/12	0/12
Marei et al., 2021 [15]	8 mg gentamicin in 20 mL 0.9% NaCl or 20 mg Gent in 50 mL 0.9% NaCl (per bladder capacity)	3 years	11/19	1	ND	One detectable	ND	1/24

* Sodium Chloride (NaCl). ** Successful outcome defined as the prevention of UTI, a ≥ 50% reduction in the rate of UTI, eradication of antimicrobial-resistant organisms, or the change in sensitivities allowing for the oral antimicrobial treatment of UTI.

**Table 4 jcm-11-05703-t004:** Administration, outcomes, and safety of use of an intravesical aminoglycoside in combination with polymyxin for UTI.

Study	Intervention	Follow-Up	Successful Outcome **	Discontinued Treatment	Side Effects	Serum Level	Change in Sensitivities	Developed New Resistance
Pearman et al., 1988 [16]	150 mg kanamycin + 30 mg colistin in 25 mL sterile water	130 days	2/7	ND	0	ND	ND	ND
Pearman 1979 [17]	kanamycin 150 mg + colistin 30 mg in 25 mL sterile water	120 days	9/17	ND	ND	ND	ND	ND
Linsenmeyer et al., 1998 [18]	30 mL neomycin/polymyxin solution	6 months	9/12	2/12	Allergy	ND	9/12	3/12
Anderson 1980 [19]	30 mL 160 mg neomycin, 800,000 polymyxin-B in sterile water	ND	17/17	ND	ND	ND	ND	ND
Waites et al., 2006 [20]	30 of 40 mg/mL neomycin sulfate and 200,000 units/mL polymyxin B	8 weeks	23/30	8/30	Autonomic dysreflexia (*n* = 2)	ND	12/30	18/30
Rhame and Perkash 1979 [21]	50 mL of 120 ug/mL neomycin and 60 ug/mL polymyxin B	28 months	51/70	ND	ND	ND	1/70	ND
Huen et al., 2019 [22]	30–50 mL 480 mg gentamicin in 1 L 0.9% NaCl	6 months	52/52	6/52	ND	ND	14/52	0

** Successful outcome defined as the prevention of UTI, a ≥50% reduction in the rate of UTI, eradication of antimicrobial-resistant organisms, or the change in sensitivities allowing for the oral antimicrobial treatment of UTI.

## Data Availability

Not applicable.

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
