# Peer review of "Are Intravesical Aminoglycosides the New Gold Standard in the Management of Refractory Urinary Tract Infection: A Systematic Review of Literature"

_jcm, 2022, doi:10.3390/jcm11195703_

Round 1
Reviewer 1 Report
This study is a narrative review of intravesical treatment with aminoglycosides for the treatment of refractory UTI. It seems that the literature search was through and the review was done properly according to the Cochran standards for systemic reviews.
The main problem is that there are not many studied on this topic, the articles reviewed were all retrospective and in some the number of patients was very small. In addition, it is not clear if the definition of refractory UTI was similar in all the studies reviewed. The authors describe a mixture of treatment for refractory UTI and prophylaxis or recurrent UTI. These are different clinical problems and there needs to be a distinction between these to medical problems. Also it is not clear what is the definition of a successful outcome and if the follow-up was long enough in the studies included to rule out the recurrence of infection. In one study the follow up was only one week, in another 4 days and in some there was no follow up. These issues need to be described and discussed as they are critical for the assessment of success or failure of the treatment. The short follow up in some of the studies prevents the assessment of success or failure of the treatment.
Author Response
Reviewer's note: This study is a narrative review of intravesical treatment with aminoglycosides for the treatment of refractory UTI. It seems that the literature search was through and the review was done properly according to the Cochran standards for systemic reviews.
Answer: We thank the reviewer for their positive and supportive comments. We agree that we have done it as per the Cochrane standards.
The main problem is that there are not many studied on this topic, the articles reviewed were all retrospective and in some the number of patients was very small. In addition, it is not clear if the definition of refractory UTI was similar in all the studies reviewed. The authors describe a mixture of treatment for refractory UTI and prophylaxis or recurrent UTI. These are different clinical problems and there needs to be a distinction between these to medical problems. Also it is not clear what is the definition of a successful outcome and if the follow-up was long enough in the studies included to rule out the recurrence of infection. In one study the follow up was only one week, in another 4 days and in some there was no follow up. These issues need to be described and discussed as they are critical for the assessment of success or failure of the treatment. The short follow up in some of the studies prevents the assessment of success or failure of the treatment.
Answer: We agree with the criticisms regarding the shortcoming and the limitations of the review. We have added this under the limitations section of the discussion, which reads –
The definition of refractory UTI and that of successful treatment was not consistent in studies, which also had limited number of patients while describing treatment or prophylaxis with follow-up duration that was quite variable. The short follow-up in some studies prevents the true assessment of success in these patients.
Reviewer 2 Report
I would like to thank and congratulate the authors for such a comprehensive paper, easy to read and concise. I honestly do not see anything that should need changes, limitations and the possible weak points are honestly exposed and well discussed. Above all, a small systematic review but with fair data in it also with a clinical relevance. The papers subjected to the review cover the wide spectrum of the most clinical relevant conditions leading to refractory urine infection.
The main question or topic addressed in this systematic review is the role of IVA instillation as alternative treatment for recurrent and refractory ITUs.
This is a clinic relevant and interesting issue to research into due to the increasing multi-drugs resistant organisms, which is a severe problem that medicine has to face currently. The interest of this review lays into analysing the previous studies about this alternative therapy, which in some cases diminishes the previous antibiotic resistances.
Many studies are showing the problem faced when treating these recurrent and refractory ITUs, and many alternatives to antibiotics are being studied (i.e the polybacterial sublingual vaccine consisting of whole-cell inactivated bacteria and also other intravesical therapies like hialuronic acid etc) and this is an important matter we need to study more and invest efforts to as not in every country or health system each product will be financed or even accepted to be commercialised. This is as the authors stated a big burden in terms of economical resources and quality of life of our patients.
The paper is well written with minimal typos or grammar mistakes. Also it is completely clear and easy to read. Methods are well established and fair described, so are the results and the tables. It might be that the different groups of patients, wide spectrum of diseases as underlying pathologies and different combinations of treatments confuse some readers. But I think they already addressed this matter in the limitations part satisfactory.
The authors addressed the main question posed in the introduction. If anything should be added in this review whose clearness and simplicity made the main problem very easy to understand already, it would be to develop in the discussion and maybe mention in the introduction other intravesical therapies or the vaccine as current alternative therapies to improve the burden of antimicrobial. However the main aim of this study is to focus in the safety and efficacy of the chosen therapy by the authors: In-travesical aminoglycoside instillation; and this is made in a comprehensive and concise way. Without entering a discussion or falling into conclusions to favour any treatment in particular, they provide useful information and they expose their limitations in a coherent way regarding the success as increase in antimicrobial sensitivity and discuss the also possible increase in resistances in some cases.
More studies, reviews and follow up studies are needed to have clear conclusions, but this is just a modest review introducing a therapy and their results in different series.
Author Response
Reviewer's note: I would like to thank and congratulate the authors for such a comprehensive paper, easy to read and concise. I honestly do not see anything that should need changes, limitations and the possible weak points are honestly exposed and well discussed. Above all, a small systematic review but with fair data in it also with a clinical relevance. The papers subjected to the review cover the wide spectrum of the most clinical relevant conditions leading to refractory urine infection.
Answer: We really appreciate the reviewer’s supportive comments towards our manuscript. We again thank them as they have agreed to all of our results and discussion with no changes needed.
Reviewer's note: The main question or topic addressed in this systematic review is the role of IVA instillation as alternative treatment for recurrent and refractory ITUs. This is a clinic relevant and interesting issue to research into due to the increasing multi-drugs resistant organisms, which is a severe problem that medicine has to face currently. The interest of this review lays into analysing the previous studies about this alternative therapy, which in some cases diminishes the previous antibiotic resistances.
Answer: We again agree with the reviewer on this aspect of clinical relevance and importance of IVA.
Reviewer's note: Many studies are showing the problem faced when treating these recurrent and refractory ITUs, and many alternatives to antibiotics are being studied (i.e the polybacterial sublingual vaccine consisting of whole-cell inactivated bacteria and also other intravesical therapies like hialuronic acid etc) and this is an important matter we need to study more and invest efforts to as not in every country or health system each product will be financed or even accepted to be commercialised. This is as the authors stated a big burden in terms of economical resources and quality of life of our patients. The paper is well written with minimal typos or grammar mistakes. Also it is completely clear and easy to read. Methods are well established and fair described, so are the results and the tables. It might be that the different groups of patients, wide spectrum of diseases as underlying pathologies and different combinations of treatments confuse some readers. But I think they already addressed this matter in the limitations part satisfactory.
Answer: We agree with the reviewer and as they mention we have tried to answer to the question but have also addressed the limitations.
Reviewer's note: The authors addressed the main question posed in the introduction. If anything should be added in this review whose clearness and simplicity made the main problem very easy to understand already, it would be to develop in the discussion and maybe mention in the introduction other intravesical therapies or the vaccine as current alternative therapies to improve the burden of antimicrobial. However the main aim of this study is to focus in the safety and efficacy of the chosen therapy by the authors: In-travesical aminoglycoside instillation; and this is made in a comprehensive and concise way. Without entering a discussion or falling into conclusions to favour any treatment in particular, they provide useful information and they expose their limitations in a coherent way regarding the success as increase in antimicrobial sensitivity and discuss the also possible increase in resistances in some cases. More studies, reviews and follow up studies are needed to have clear conclusions, but this is just a modest review introducing a therapy and their results in different series.
Answer: We are thankful to the reviewer and based on their recommendation, no changes are needed to our manuscript.
Round 2
Reviewer 1 Report
I do not think that the authors have addressed my comments in a sufficient manner and have not made any changes in the way the data is presented.
Author Response
Question - This study is a narrative review of intravesical treatment with aminoglycosides for the treatment of refractory UTI. It seems that the literature search was through and the review was done properly according to the Cochran standards for systemic reviews.
Answer: We thank the reviewer for their positive and supportive comments. We agree that we have done it as per the Cochrane standards.
Question - The main problem is that there are not many studies on this topic, the articles reviewed were all retrospective and in some the number of patients was very small. In addition, it is not clear if the definition of refractory UTI was similar in all the studies reviewed. The authors describe a mixture of treatment for refractory UTI and prophylaxis or recurrent UTI.
Answer: We agree with the reviewer. We went through each paper to further investigate every definition of refractory UTIs. We have added a ‘Definition of recurrent UTI’ column to table 1 and table 2 that outlines this for each paper. We have clarified in the last paragraph of the limitations that the definition of refractory UTI was not similar in all the studies reviewed.
We have also added a paragraph under data extraction and outcome of interest which reads –
The measure of a successful outcome differed between the studies. Therefore, to aid comparison, this review has defined a successful outcome as the prevention of UTI, ≥50% reduction in the rate of UTI, eradication of antimicrobial-resistant organisms, or the change in sensitivities allowing oral antimicrobial treatment of UTI.
Question - These are different clinical problems and there needs to be a distinction between these to medical problems.
Answer: The difference between medical and clinical problems is not clearly described in the included papers. However, we have divided the tables in order to separate the co-morbidities/risk factors in tables 1 and 2 and the side effects/complications of IVA in tables 3-4. We have changed the heading of table 1 and 2 to " Additional risk factors/comorbidities " which offers more clarity.
Question - Also it is not clear what is the definition of a successful outcome and if the follow-up was long enough in the studies included to rule out the recurrence of infection
Answer: Our definition of a successful outcome is defined in the methodology (as mentioned above). The outcome parameters allowed for comparison between papers that greatly differed in their definition of a successful outcome. Adding this to the tables/ discussing the outcomes per paper as defined by their parameters of a successful outcome was confusing and detracted from any potential comparisons that could be made.
We have changed the wording in the methodology and added an Asterix addendum to the tables to further clarify this.
Question - In one study the follow up was only one week, in another 4 days and in some there was no follow up. These issues need to be described and discussed as they are critical for the assessment of success or failure of the treatment. The short follow up in some of the studies prevents the assessment of success or failure of the treatment.
Answer: We agree with the reviewer. This is one of the biggest limitations of this review. The papers are often describing very short follow up times, preventing to compare the outcomes and duration of the treatment effectiveness. The differing lengths of follow up had been already explored in the discussion and we now added a further section in the limitations which hopefully addresses this, which reads - .
“The review is limited by the heterogeneity of the studies and variation in sample sizes. Variability in study design was noted in the definition of recurrent/refractory UTI, measure of successful outcomes, categorisation of groups into treatment versus prophylactic IVA, and the duration of follow-up. As a result of these key design differences, it is not possible to accurately evaluate the merits or failures of IVA treatment amongst these patients (tables 1-2).
The short follow-up in some studies prevents the true assessment of success in these patients. Five of thirteen papers are characterized by short follow up duration. Haldorson et al, Wan et al, Praba et al, Cox and Waites et al describe a follow up that goes from a minimum of 4 days to a maximum of 8 weeks. Given that the definition of recurrent UTIs is ≥2 infections in six months or ≥3 infections in one year, UTIs resolution should be defined with a longer follow-up after the treatment. However, in view of limited patient numbers and poor patient compliance to long term follow up, these papers were also included.
Future studies with more robust methodology and longer follow up are required to draw meaningful conclusions. This would allow for a more accurate assessment of treatment efficacy with a view to aid clinical guidelines.” It was noted the patient demographics were often not disclosed or sparsely documented in the included papers. This could be explored further to evaluate the optimum demographic characteristics of patients that would benefit from IVA and evaluate the impact of patient co-morbidity.
